



# Probing environmental and tectonic changes underneath Ciudad de México with the urban seismic field

Laura Ermert[1, **], Enrique Cabral Cano[2, *], Estelle Chaussard[3, *], Darío Solano Rojas[4, *],
Luis Quintanar[2, *], Diana Morales Padilla[4], Enrique A. Fernández Torres[2], and Marine A. Denolle[1]

[1]Department of Earth and Space Sciences, University of Washington, Seattle WA, USA
[**]Now at: Swiss Seismological Service, ETH Zürich, Zürich, Switzerland
[3]Independent Researcher
[2]Instituto de Geofísica, Universidad Nacional Autónoma de México, CDMX 04510, Mexico
[4]Facultad de Ingeniería, Universidad Nacional Autónoma de México, CDMX 04510, México
[*]Who contributed equally to this work

**Correspondence:** Laura Ermert (laura.ermert@sed.ethz.ch)

**Abstract.** The subsurface materials of Ciudad de México have unique mechanical properties that give rise to strong site effects. We investigated temporal changes in the seismic velocity at strong-motion and broad-band seismic stations throughout Mexico City, including sites with different geologic characteristics ranging from city center locations situated on lacustrine clay to hillsite locations on volcanic bedrock. We used autocorrelations of urban seismic noise, enhanced by waveform clustering, to

extract subtle seismic velocity changes by coda wave interferometry. We observed and modeled seasonal, co-, and postseismic changes, as well as a long-term linear trend in seismic velocity. Seasonal variations can be explained by self-consistent models of thermo-elastic and poro-elastic changes in the subsurface shear wave velocity. Overall, sites on lacustrine clay-rich sediments appear to be more sensitive to seasonal surface temperature changes, whereas sites on alluvial and volcaniclastic sediments and on bedrock are sensitive to precipitation. The 2017 $M_w$ 7.1 Puebla and 2020 $M_w$ 7.4 Oaxaca earthquakes both caused

a clear drop in seismic velocity followed by a time-logarithmic recovery that may still be ongoing for the 2017 event at several sites, or that may remain incomplete. The slope of the linear trend in seismic velocity is correlated with the downward vertical displacement of the ground measured by Interferometric Synthetic Aperture Radar, suggesting a causative relationship and supporting earlier studies on changes in the resonance frequency of sites in the Mexico City basin due to groundwater extraction. Our findings show how sensitively shallow seismic velocity, and in consequence, site effects, react to environmental,

tectonic and anthropogenic processes. They also demonstrate that urban strong-motion stations provide useful data for coda-wave monitoring given sufficiently high-amplitude urban seismic noise.





## 1 Introduction

Near-surface geological structure and soil properties are important determinants of seismic hazard (e.g. Field, 2000). Shal-
low, poorly consolidated sediments can strongly amplify long-period seismic waves, and the strong impedance contrast to the
underlying bedrock can trap energy and vastly prolong ground motion (e.g. Roten et al., 2008; Cruz-Atienza et al., 2016).
Therefore, considerable effort is invested to determine shallow sediment properties in urban areas (see Foti et al., 2019, for
a review). One of the decisive quantities controlling site response is the shallow shear wave velocity, which is also used as
a hazard assessment parameter in the form of $v_{s30}$, or shear wave velocity averaged in the top 30 m depth. Shallow seismic
velocities react strongly to environmental variations, as has been documented by time-lapse imaging (Bergamo et al., 2016) and
ambient noise monitoring (e.g. Steinmann et al., 2021). Ambient noise monitoring is based on comparing short-term interfer-
ometric measurements like cross-correlation or deconvolution of continous seismic data to a long-term reference (Obermann
and Hillers, 2019). Under the assumption that the coda of the resulting waveforms is predominantly sensitive to changes in the
elastic medium, one can measure subtle relative advances and delays in the current waveform compared to the reference, which
are approximately linearly related to relative velocity changes $\frac{dv}{v}$. Using this technique reveals that seismic velocity varies
with groundwater level (Sens-Schönfelder and Wegler, 2006; Lecocq et al., 2017; Fokker et al., 2021; Rodríguez Tribaldos and
Ajo-Franklin, 2021), precipitation, soil moisture and snow load (Obermann et al., 2014; Voisin et al., 2016; Wang et al., 2017;
Donaldson et al., 2019; Andajani et al., 2020; Mordret et al., 2020; Feng et al., 2021; Illien et al., 2021), ground temperature
(Richter et al., 2014), thawing of the permafrost (Ajo-Franklin et al., 2017; James et al., 2019; Lindner et al., 2021), and even
centimeter-scale layers of soil freezing (Steinmann et al., 2021). Droughts can induce longer-term changes of seismic velocity
(Clements and Denolle, 2018; Mao et al., 2021), as can soil compaction (Taira et al., 2018). Although the reported changes are
usually small, on the order of approximately 1 % peak-to-peak amplitude or 0.01 – 0.1 %/year trend, they clearly show that
shallow sediment properties are time-dependent.

A second phenomenon relevant to site response is the non-linear behavior of soft near-surface sediments subject to large dy-
namic strains, including shear modulus reduction and plastic deformation (Wu and Peng, 2012; Oral et al., 2019; Bonilla et al.,
2019). Numerous recent studies utilizing ambient noise interferometry have reported temporary velocity drops consistent with
shear modulus reduction during the shaking from moderate to large earthquakes, generally followed by post-seismic relaxation
that is approximately linear with the logarithm of time (e.g. Brenguier et al., 2008b; Hadziioannou et al., 2011; Wu and Peng,
2012; Froment et al., 2013; Hobiger et al., 2016; Wang et al., 2017). Most studies found small velocity drops on the order of 0.1
– 1 % following ground shaking. However, much larger reductions on the order of 1 – 10 % and more were reported in studies
which explicitly targeted shallow structure on the order of 100 m below the surface, as would be expected for non-linear elas-
ticity (Nakata and Snieder, 2011; Viens et al., 2018). The velocity reduction is reported to be even stronger during the shaking
itself (Bonilla et al., 2019; Bonilla and Ben-Zion, 2020); however, most studies lack sufficient time resolution to capture this
short-lived effect.

In the present study, we investigate both linear and non-linear changes of the seismic velocity underneath Mexico City. Mexico
City has suffered devastating ground shaking not least due to the response of lacustrine clay deposits in the center of the Mexico



City basin (Anderson et al., 1986; Singh et al., 1988a; Sahakian et al., 2018; Arciniega-Ceballos et al., 2018; Cruz-Atienza et al., 2016), and continues to face high seismic hazard. Mexico City is also affected by extremely rapid ground subsidence (Cabral-Cano et al., 2008; Chaussard et al., 2021). The main motivation for our study is to understand how environmental factors and strong ground motions from earthquakes influence the seismic velocity structure of the shallow to intermediate sediments. Furthermore, we aim to demonstrate that it is possible to monitor such temporal changes using data from continuous recordings of urban seismic noise at a relatively sparse strong motion sensor network with seismic interferometry. We characterize the observed velocity changes through physics-based modeling and probabilistic inversion of key parameters like velocity drop and sensitivity to surface temperature.

In the following, we describe the data (sect. 2) and the processing approach we used to overcome the particular challenges of urban seismic noise (Groos and Ritter, 2009; Schippkus et al., 2020, sect. 3). In sect. 4, we introduce our model for velocity changes and the probabilistic inversion of model parameters. Finally, we present and discuss the observations and inversions (sect. 5), and conclude with an outlook and recommendations for further research (sect. 6).

## 2 Study area and data

The National Seismological Service of Mexico operates a state-of-the-art seismic network in the Valley of Mexico, the Red Sismica del Valle de México (RSVM) (Quintanar et al., 2018). Here, we use data from RSVM strong-motion sensors that were mostly installed in 2017. Strong-motion sensors are chosen for dense deployment in some urban areas with high seismic hazard and thus provide a valuable data source for ambient noise monitoring, despite their comparably low sensitivity (e.g. Tokyo metropolitan area, Seattle, Berkeley Borehole network in the San Francisco Bay area; Kasahara et al., 2009; Viens et al., 2018; Bonilla et al., 2019). We first compare results obtained from a co-located seismometer and accelerometer to verify that the urban noise at periods of 2 seconds and shorter is well captured by both types of sensors and that results are consistent. We then focus on continuous strong-motion recordings at 12 locations representative of the different site conditions in Mexico City, including station locations on soft, intermediate, and hard sites as defined by Quintanar et al. (2018). We supplement these with broad-band sensor observations from the Geoscope network site G.UNM and four stations of the temporary Tectonic Observatory deployment (MASE, 2007).

The station G.UNM includes a co-located STS-1 seismometer (broadband sensor) and Kinemetrics EpiSensor accelerometer (strong motion sensor), and has been recording continuously since 1995 for the broadband sensor and since 2013 for the strong motion sensor on the main campus of the National Autonomous University of Mexico (green triangle in Figure 1). The upper panel of Figure 2 shows a spectrogram obtained from the autocorrelations of the North component of the broadband seismometer between 1995 and 2021. It illustrates that urban noise levels around 1 Hz surpass Peterson's New High Noise Model of -120dB (Peterson, 1993). Furthermore, instrumental effects are clearly visible, such as the change in instrument gain during 2008. The recorded urban noise is not stationary; a group of short-lived spectral peaks appears only in 2015. In 2020, the noise level drops at the onset of the Covid-19 pandemic (see Pérez-Campos et al., 2021). We will later on compare observed and modeled results from the two co-located sensors to assess the use of strong-motion stations for coda-wave monitoring (see



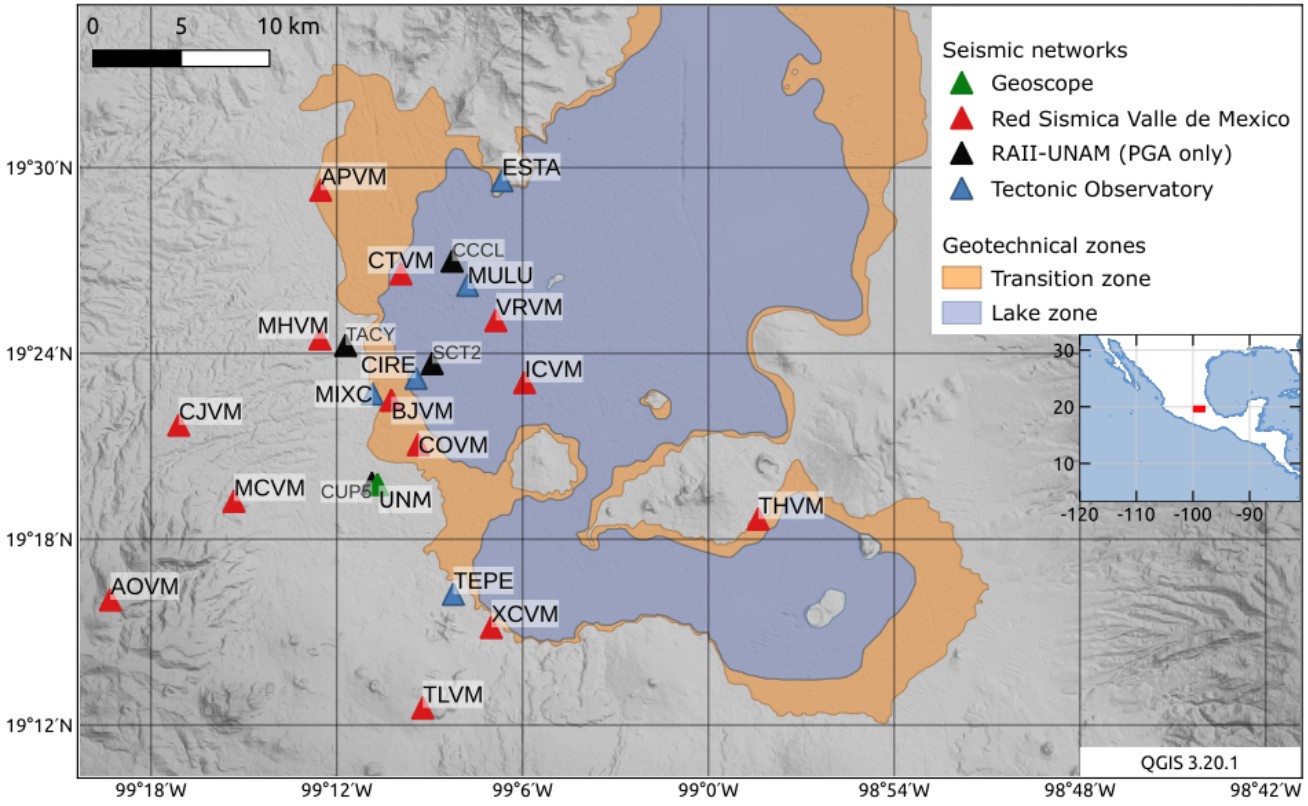

**Figure 1.** Shaded relief (Farr et al., 2007), geotechnical zonation (Gobierno de la Ciudad de México, 2017), and location of seismic stations in the Valley of Mexico region (Quintanar et al., 2018; MASE, 2007; Roult et al., 2010). Lake zone (blue outline) and transition zone (orange outline) include locations where the shallow subsurface is characterized by quaternary lacustrine sediment. Strong motion and broad-band sensors at Geoscope station G.UNM (green triangle) were used for comparing velocity changes assessed with different instruments, and for analysis of the velocity changes. Strong motion sensors of the Red Sísmica del Valle de México (red triangles) and broad-band stations of the temporary Tectonic Observatory (TO) deployment (blue triangles) and G.UNM were used for analysis of velocity changes. Strong-motion stations of the Red Acelerográfica del Instituto de Ingeniería (RAII-UNAM, black triangles) were used for determining additional peak ground acceleration values during the 2017 $M_w$7.1 Puebla earthquake.

sect. 5.1).

For analyzing velocity changes, we focus on urban seismic noise autocorrelations. Autocorrelations have been successfully used for monitoring earthquake damage and climate effects on the near-surface velocities in previous studies (e.g. De Plaen et al., 2019; Sánchez-Pastor et al., 2019; Feng et al., 2021). Moreover, coherency of the ambient noise between stations is poor



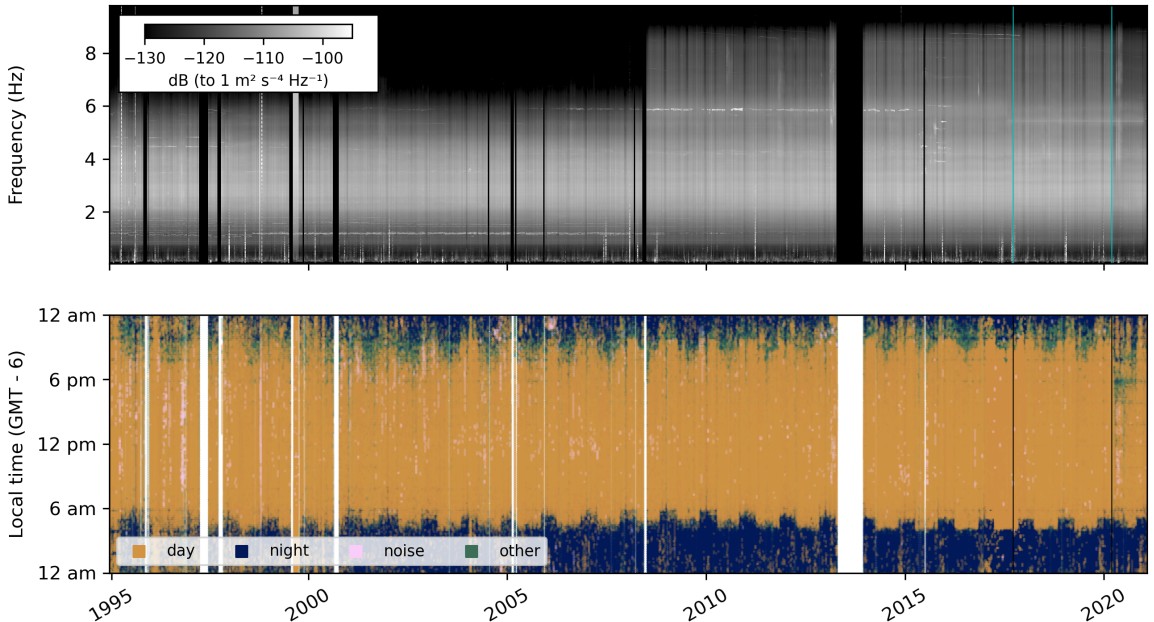

**Figure 2.** Top: Acceleration spectrogram of urban noise at G.UNM (North component), broadband seismometer. Prior to June 2008, the sensor was operated at lower gain. The spectrogram illustrates high noise levels characteristic for the urban location. Faint changes in the spectrum coincide with the September 2017 earthquake (left cyan line), as well as a marked drop in noise level during the Covid-19 pandemic (right cyan line: First announcement of anti-pandemic measures). Bottom: Clustering of short-term N-N autocorrelation waveforms in the frequency band 2-4 Hz. Clustering results suggest that autocorrelation waveforms change on a day-night rhythm, a weekly rhythm (not visible here), and an annual switch to and from daylight saving time. Similar urban patterns appear in clustering results at all stations, mostly at frequencies $\geq 0.5$ Hz. Black vertical lines indicate 2017 earthquake and announcement of anti-pandemic measures. The color scale was chosen for accessibility (Crameri et al., 2020).

at frequencies above 1 Hz, which may be due to the near-station sources of high urban noise, the strong attenuation in the basin
sediment (e.g. Cruz-Atienza et al., 2016), and the comparably low sensor sensitivity.

## 3   Measuring velocity changes in an urban setting

We remove large global earthquakes of $M_w = 7$ and above according to the global CMT catalogue using magnitude-dependent window lengths following the approach of Ekström (2001). Local earthquakes in a $5°$ radius from the ISC catalog (International Seismological Centre, 2022; Bondár and Storchak, 2011) are removed by cutting out a window starting ten seconds
before the direct P-wave using iasp91 (Kennet, 1991) and ending 60 seconds after a $1 \frac{km}{s}$ surface wave. We cut data into 8-hour




segments, detrend, taper, pre-bandpass-filter them, and remove the instrument responses to correct the data to ground acceleration (including the seismometer for direct comparison of the results). Finally, we correlate 20-minute windows of the data overlapping by 10 minutes. We save all single-window correlations for further processing in hdf5 format (Folk et al., 2011). The pre-processing and correlation are performed with python module `ants`, which uses `obspy` (Beyreuther et al., 2010;

Krischer et al., 2015) for instrument correction, filtering, and other seismic data processing tasks (see sect. 6 for details on code availability).

### 3.1  Clustering correlation waveforms and selective stacking

We adopt a novel processing strategy for ambient noise correlations proposed by Viens and Iwata (2020). It is based the premise that variable incident ambient noise conditions result in different correlation waveforms. By grouping single-window

autocorrelations into clusters and stacking them selectively, we aim to increase their temporal coherence, and reduce the effect of temporally varying noise sources. Clustering is performed by applying Gaussian Mixture Models after reducing the data dimensionality through Principal Component Analysis (PCA), and the Bayesian Information Criterion (BIC) is used to determine the optimal number of clusters, balancing misfit and model complexity (Viens and Iwata, 2020).

We modify the original approach in several ways in order to adapt it to long-term urban noise. Details and rationales are

provided in the supplement. The modifications can be summarized as follows: (i) We apply clustering per octave frequency band to account for narrow-band cultural sources, (ii) we pre-determine the principal component axes on a subset of the data due to the large data volume, (iii) we normalize by dividing each waveform by its maximum absolute value, (iv) we fix the number of clusters *a-priori* at an optimum determined in a preliminary clustering run, here 4 clusters, and (v) we rely on cultural patterns with respect to local time to label the clusters as "day", "night", "noise" and "other". The result for the North-

North component of the seismometer at G.UNM is shown in Fig. 2. The urban day-night rhythm emerges for all stations at frequencies of 0.5 Hz and above, as well as for several stations inside the basin for frequencies between 0.25 Hz and 0.5 Hz. We tested the effect of using the amplitude-unbiased phase cross-correlation (Schimmel et al., 2011; Sánchez-Pastor et al., 2018) on clustering and found that although the clustering results change, the day-night rhythm is still clearly visible (Suppl. Fig. S1). This suggests that processing schemes aimed at equalizing noise amplitudes fail to suppress the effect of temporally

varying noise sources in an urban environment. We therefore use the clustering approach for the subsequent analysis. We stack the short-term correlations for the daytime clusters, which are the largest in numbers and the most consistent with time, over a duration of 10 days. We handle the short-term correlation data, filtering, clustering, and stacking with the python module `ruido` (see sect. 6).

### 3.2  Stretching measurement

We use the stretching method to measure relative changes in velocities (e.g. Sens-Schönfelder and Wegler, 2006). A preliminary comparison to the moving window cross-spectral method (e.g. Takano et al., 2020) showed good overall agreement. We use a multiple reference approach due to the lack of long-term waveform coherence in our observations; details are provided in the Supplement. Multiple reference approaches have been previously used (in somewhat different forms), e.g. by Sens-Schönfelder





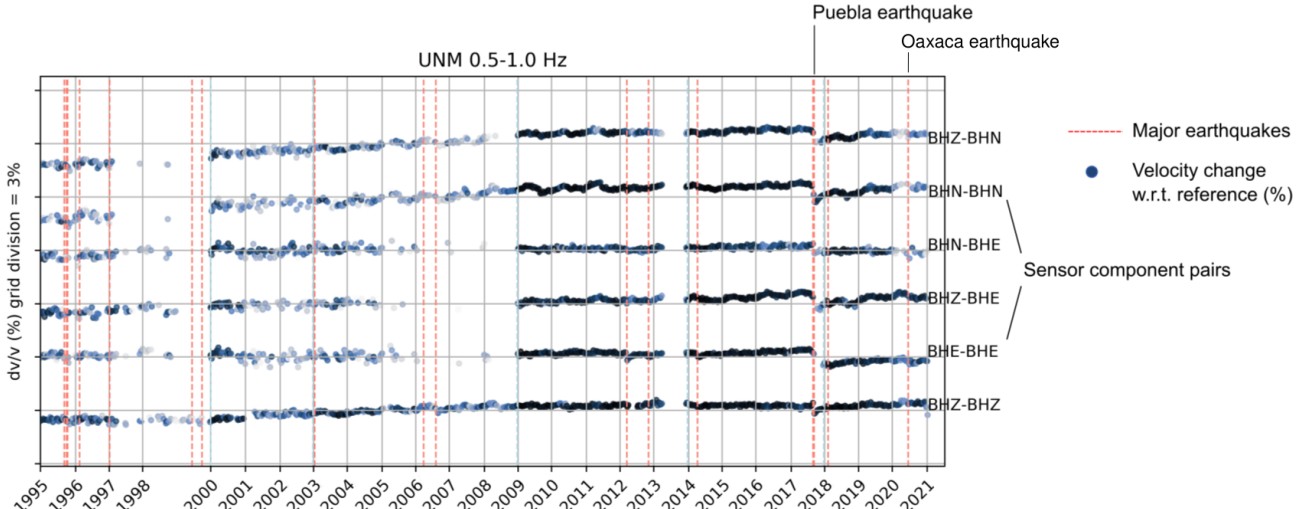

**Figure 3.** Velocity changes at station G.UNM, all unique channel pairs, 16-40 seconds lag. Channels labelled BHE, BHN and BHZ are the East, North and vertical channels recorded by the seismometer. Color hue shows the correlation coefficient of the current and reference trace after stretching, with light colors indicating low and dark colors high correlation.

et al. (2014) and Donaldson et al. (2019). The stretching is performed in four frequency bands (0.5-1, 1-2, 2-4 and 4-8 Hz), and

in two windows on each stack, one of which extends from 4 to 10 times the longest period of interest (defined by the frequency band) and the other from 8 to 20 times the longest period of interest (e.g. for a 0.5-1 Hz observation, we measure velocity changes in the 8-16 seconds and 16-40 seconds windows). Figure 3 shows results at station G.UNM, 0.5-1 Hz, using the coda window at 16-40 seconds. We note that the information contained in the single station cross-correlations (mixed components) is visually consistent with the pure autocorrelations (Fig. 3). We thus proceed with the pure autocorrelations, mostly because their

interpretation is conceptually simpler; this also reduces computational requirements for the following inversion. We also note that data quality as measured by the correlation coefficient between reference trace and current trace after stretching (CC_best) (shown by color hue) increases markedly after the sensor at G.UNM was set to high-gain mode in 2008. For further analysis, we discard data points with CC_best < 0.6.

## 4 Modeling velocity changes

Visual inspection of the time series in Figure 3 suggests that they contain different patterns: seasonal changes, a co-seismic drop coinciding with the 2017 $M_w$ 7.1 Puebla and 2020 $M_w$ 7.4 Oaxaca earthquakes, as well as post-seismic recovery. We model these processes by considering the influence of surface temperature, precipitation, earthquake shaking and healing on the seismic velocity. In addition, as we note that the seismic velocity generally increases over time, we include a linear trend. For simplicity, we sum the submodels as a linear combination (see also Hobiger et al., 2014, 2016; Wang et al., 2017; Donaldson





et al., 2019; Feng et al., 2021), although there may be interactions that lead to non-linearity, e.g. between water content and temperature (Sens-Schönfelder and Eulenfeld, 2019). We use a Markov Chain Monte Carlo (MCMC) inversion to infer the unknown model parameters. We assume that the changes observed at the surface are dominated by changes in Rayleigh-wave phase velocity, which are mainly sensitive to changes in shear wave velocity at depth, and mostly insensitive to changes in P-wave velocity (see sect. 5.2).

## 4.1 Earthquake effects

The 2017 $M_w$ 7.1 Puebla earthquake caused strong ground motion in Mexico City reaching peak ground accelerations over 15% g (Alberto et al., 2018) and a sharp acceleration of subsidence in various locations (Solano-Rojas et al., 2020). We observe velocity drops coinciding with this earthquake followed by approximately logarithmic recovery at most stations. More subtly, a similar signal is observed at several stations following the 2020 $M_w$ 7.4 Oaxaca (Sta. María Xadani) earthquake, see Fig.

5. Physical processes causing the co-seismic velocity drop may be caused by plastic deformation (fractures opening due to shaking-induced dynamic stresses that exceed the Earth material yield strength (Gassenmeier et al., 2016; Bonilla et al., 2019)) or by a non-linear mesoscopic elasticity that describes the loss and reestablishment of chemical bonds or capillary bridges that change frictional contacts (Sens-Schönfelder et al., 2018)). Snieder et al. (2017) proposed a model for the long-term post-seismic recovery described as the superposition of exponential relaxation mechanisms with different relaxation timescales $\tau$

ranging from $\tau_{min}$ to $\tau_{max}$ after the time of the earthquake, $t_{quake}$:

$$\frac{c(t) - c_0}{c_0} = \frac{s}{c_0} \ln\left(\frac{\tau_{max}}{\tau_{min}}\right)^{-1} \int_{\tau_{min}}^{\tau_{max}} \frac{1}{\tau} \exp(-t/\tau)\,d\tau \qquad \text{for } t \geq t_{quake}. \tag{1}$$

Here, $s$ is a negative value indicating the drop of seismic velocity from its previous value $c_0$. For intermediate timescales, $(\tau_{min} \ll t \ll \tau_{max})$, this model captures the approximately logarithmic time dependence, while it tends to $c_0$ for large times, and is finite for $t = 0$ where a purely logarithmic recovery is not defined. Hypothesizing that the minimum timescale $\tau_{min}$ is

below the temporal resolution of our measurements, we set $\tau_{min} = 0.1$ s and fit both $\tau_{max}$ and the co-seismic step drop $s$ through the inversion described below. Velocity changes due to non-linear elasticity are depth-dependent (e.g. Wang et al., 2019). For the earthquake-induced non-linearity, we account for depth-dependence only insofar as we invert data from different frequency bands separately (Obermann et al., 2014; Sawazaki et al., 2016; Wu et al., 2016).

## 4.2 Seasonal effects

The following sections describe how we model the effects of surface temperature (4.2.2) and precipitation (4.2.3) on Rayleigh-wave phase velocity. In both cases, we use analytical solutions to diffusion-like problems for a homogeneous half-space to model their effect on $v_s$ from the surface to 2 km depth in terms of a) thermo-elastic stress and b) pore pressure. In particular, we use a) the solution of Berger (1975) as formulated by Richter et al. (2014) and b) the solution of Roeloffs (1988) as formulated by Talwani et al. (2007). In the thermo-elastic model, annual and sub-annual periodic surface temperature changes

diffuse through the shallow subsurface. In the pore pressure model, rain leads to a sudden increase in pore pressure near the





surface, which then diffuses towards depth. We compute surface-wave sensitivity kernels with `surf`, a python package based on the Takeuchi and Saito (1972) solutions to the surface wave eigenproblem in layered, anisotropic elastic media (Fichtner, 2020). This is done using station-specific 1-D velocity profiles and assuming an isotropic medium (see sect. 4.2.1). Finally, we integrate the depth-dependent $v_s$-change to obtain the predicted surface-wave phase velocity ($c$) changes.

The solutions at depth in the homogeneous halfspace depend on a) thermal diffusivity and b) hydraulic diffusivity of the sediments, which are not well known and in the case of b) can vary by several orders of magnitude (Roeloffs, 1996). Inverting for these parameters probabilistically would be challenging, because it requires re-evaluating the diffusion terms at all depths for each iteration. Instead, we run the inversion repeatedly for 6 homogeneous half-spaces characterized by a) 3 and b) 2 trial diffusivities in the ranges a) 0.15 to 2 mm$^2$/s (Richter et al., 2014) and b) 0.0001 to 4 m$^2$/s (Roeloffs, 1996). We retain the

diffusivity for each site that produces the best fit across all components and frequency bands.

### 4.2.1   Site-specific velocity profiles

We need estimates of shear wave velocity $v_s$, compressional wave velocity $v_p$, and density $\rho$ to establish the effect of velocity changes at depth on surface-wave velocity changes $\frac{dc}{c}$. To account for the different subsurface geology at each station, we use a classification as "hard", "intermediate", and "soft" provided by Quintanar et al. (2018) and we consider the aquitard thickness

information from Solano-Rojas et al. (2015). We then construct the 1-D profile of $v_s$, $v_p$, and $\rho$ at each site as follows:

1. Assign a depth to bedrock value: Hard sites 0 m, intermediate sites 100 m, and soft sites 300 m, coarsely representing the lower boundary of volcanic and alluvial sediments, which generally deepens towards the center of the basin.

2. Assign a site-dependent lacustrine sediment thickness ("clay") to the intermediate and soft sites derived from the thickness of the upper aquitard (Solano-Rojas et al., 2015).


3. Assign velocities to clays, volcanic and alluvial sediments, and bedrock according to table 1, where sediment 1 is the upper half of the sediment column (alluvial), sediment 2 is the lower half (volcanic sediments), bedrock 1 is considered up to 1000 m below the sediment, and bedrock 2 at greater depths.

The values for geologic structure and seismic properties are based on a synopsis of the work of Pérez Cruz (1988); Chavez-Garcia and Bard (1994); Singh et al. (1995, 1997) and Shapiro et al. (2001), which serve as references for modeling the seismic

velocity structure of the Mexico City basin (Cruz-Atienza et al., 2016; Asimaki et al., 2020). The velocity values suggest that Poisson's ratio approaches 0.5 for the lacustrine sediments. Supplementary Fig. S2 shows that our rule-based velocity model is in reasonable agreement with the shear wave velocity profiles based on well logs from Singh et al. (1995, 1997). Future investigations would benefit from a more detailed 3-D velocity model of the basin and surroundings.

    As the surface-wave sensitivity kernels in Supplementary Fig. S3 illustrate, the model leads to the following behaviors. At

hard sites, the sensitivity is spread over the shallowest 1000 m and the peak sensitivity shifts upwards with frequency from about 450 m in the lowest to about 50 m in the highest frequency band. At soft sites, the sensitivity is concentrated in the shallowest 50 m for the lowest and the shallowest 5 m for the highest frequency band, but in any case inside the low-velocity





| Unit | $v_s$ (m/s) | $v_p$ (m/s) | $\rho$ (kg/m$^3$) |
|---|---|---|---|
| Clay | 50 | 800 | 1250 |
| Sediment 1 | 400 | 2500 | 2000 |
| Sediment 2 | 800 | 2500 | 2000 |
| Bedrock 1 | 1050 | 2600 | 2000 |
| Bedrock 3 | 2100 | 3600 | 2000 |

**Table 1.** Compilation of approximate elastic properties based on the synopsis of Pérez Cruz (1988); Singh et al. (1995, 1997); Shapiro et al. (2001).

sediment. At intermediate sites, the behavior is similar to hard sites for the lower frequencies up to 2 Hz and similar to soft sites for the higher frequencies. Our single-station analysis presumes a laterally homogeneous structure near each station. While this is a rather crude assumption, we believe that it captures important aspects of the basin, in particular the very shallow sensitivity of our observations at soft sites.

### 4.2.2 Surface temperature effects

We adopt the approach of Richter et al. (2014) to compute the thermo-elastic stress, neglecting variations at greater depth. The shear wave velocity change depends linearly on the temperature $T$ at location $x$, depth $z$ and time $t$,

$$\frac{\delta v_s}{v_s}(x,z,t) = s_t \cdot T(x,z,t), \tag{2}$$

(see eq. 14 of Richter et al., 2014), where we summarize material parameters into a depth-independent temperature sensitivity:

$$s_t = 2b\alpha \frac{\partial \rho v^2}{\partial \sigma_c}, \tag{3}$$

(eq. 12 of Richter et al., 2014), where $b = \frac{1+\nu}{1-\nu}$ for S-waves, $\frac{\partial \rho v^2}{\partial \sigma_c}$ describes the change of shear modulus with respect to stress, i.e. the non-linear elastic rheology, $\nu$ is Poisson's ratio, and $\alpha$ the linear thermal expansion coefficient. The single factors of this sensitivity are not particularly well known, so we fit the product $s_t$ during the inversion described below. We use surface temperature measured at the meteorologic network of the Universidad Nacional Autónoma de México (UNAM) (Instituto de Ciencias de la Atmósfera y Cambio Climatico, 2022). We expand the temperature curves into Fourier series with five terms using the nearest available meteorologic station to each seismic station. This proved important for the model fit, as the temperature variations with sub-annual period affect $\frac{dc}{c}$ . Finally, we convert the shear-wave velocity change at depth to $\frac{dc}{c}$ using the phase velocity sensitivity kernels for the fundamental-mode Rayleigh waves. During the inversions we use 3 trial values of thermal diffusivity $\kappa_t$ in the range of $1.5 \cdot 10^{-7}$ to $2 \cdot 10^{-6}$ m$^2$/s and select the $\kappa_t$ at each station that produces the minimum average misfit over all components and frequency bands.



### 4.2.3 Hydrological effects

Quantitative interpretation of observed velocity changes in terms of hydrology is a matter of current research and can be highly site-specific; it requires measuring or estimating the hydrologic changes (usually pore pressure), and estimating the sensitivity of the observed velocity change to these. We compile an exemplary, non-exhaustive selection of recent approaches in table 2. Here, we estimate the pore pressure changes from precipitation data, use a depth-varying medium response to stress based on granular media theory, and translate changes in $v_s$ to $\frac{dc}{c}$ with surface wave sensitivity kernels.

We compute the pore pressure change using the impulse response to hydraulic head change derived by Roeloffs (1988) for a homogeneous halfspace, assuming test values for hydraulic diffusivity $\kappa_h$ of $1 \text{ m}^2/\text{s}$ representing relatively impermeable unconsolidated sandy sediments and $10^{-4} \text{ m}^2/\text{s}$ representing permeable unconsolidated clay sediments (Roeloffs, 1996). Note that the sensitivity of the modelled $\frac{dc}{c}$ to hydraulic diffusivity is low because of the depth integration for surface wave phase velocity change and the high values of Poisson's ratio at the sites (Supplementary Fig. S4). We estimate the change in $v_s$ due

to pore pressure changes using granular media theory (Mavko et al., 2020). Saul and Lumley (2013) formulated the effective moduli for effective pressure $p_{\text{eff}}$, based on which:

$$\frac{1}{v_s}\frac{\partial v_s}{\partial p_{\text{eff}}} = \frac{1}{6\,p_{\text{eff}}}, \tag{4}$$

(see also Takano et al., 2017), where $p_{\text{eff}} = P_{ov} - nP_p$ with overburden pressure $P_{ov}$ and pore pressure $P_p$. We estimate overburden pressure as lithostatic pressure from the 1-D density models described in sect. 4.2.1, and unperturbed pore pressure

as hydrostatic pressure, using porosities of 0.6 and 0.2 for clay and any other sediment, respectively, following Ortega and Farvolden (1989), and assuming $n = 1$. Granular media theory was previously used by Rodríguez Tribaldos and Ajo-Franklin (2021) to model velocity changes in an unconfined aquifer. Not all sites in our study have the same hydrogeological properties, but Takano et al. (2017) found that granular medium theory can approximately explain observations of stress sensitivity over a large range of depths and geologic materials. Equation 4 tends to infinity for effective pressures approaching zero. We

mitigate this by introducing a minimal effective pressure $p_0$ as a parameter in the inversion, i.e., $p_{\text{eff}} = \max(p_0, P_{ov} - nP_p)$. The resulting $\frac{dc}{c}$ is strongly sensitive to $p_0$.

### 4.3 Probabilistic inversion

Considering the superposition of the temperature, precipitation, earthquake, and linear trend, we aim to model the phase velocity change $\frac{dc}{c}$ as

$$\frac{dc}{c} = f_{\text{temp}}(\kappa_t, s_t) + f_{\text{rain}}(\kappa_h, p_0) + f_{\text{seismic}}(\Delta c, \tau_{\text{max}}) + f_{\text{lin}}(a, b), \tag{5}$$

where $\kappa_t, \kappa_h$ are the thermal and hydraulic conductivity, respectively, $s_t$ is the sensitivity of the shear wave velocity to thermoelastic stress, $p_0$ is the minimal overburden pressure, $\Delta c$ is the co-seismic drop in observed velocity (which we assume is surface wave phase velocity), $\tau_{\text{max}}$ is the maximum timescale of exponential recovery in the slow-dynamics model of Snieder et al. (2017), and $a, b$ are the slope and offset of the linear trend. Values for the remaining variables are taken from literature.





| Publication | Wave type | Hydrological time series | Response of medium | Depth sensitivity |
|---|---|---|---|---|
| SW06[1] | Body | GWL from rain, Baseflow | $\delta v$ if $\text{GWL} < \text{GWL}_{\text{ref}}$, $-\delta v$ if $\text{GWL} > \text{GWL}_{\text{ref}}$ | Scattering kernel |
| L17 | Surface | GWL from GR4J | $v_s \propto p_p$ | Rayleigh wave $K_{vs}$ |
| W17 | Surface | $p_p$ from rain (Roeloffs, 1988) | $v_s \propto p_p$ at fixed depth | n/a (fixed depth) |
| RT21 | Surface | $p_p$ from river stage | Stress-dependent $v_s$ using granular medium theory | Rayleigh wave $K_{vs}$ |
| F21 | Surface | Measured hydraulic head | Stress-dependent $v_s$ using $\frac{\partial v_s}{\partial p_p}$ from data | Rayleigh wave $K_{vs}$ |
| **This study** | Surface | $p_p$ from rain (Roeloffs, 1988) | Stress-dependent $v_s$ using granular medium theory | Rayleigh wave $K_{vs}$ |

**Table 2.** Comparison of selected approaches to model hydrological effects on seismic velocity (not an exhaustive list). SW6, Sens-Schönfelder and Wegler (2006); L17, Lecocq et al. (2017); W17, Wang et al. (2017); RT21, Rodríguez Tribaldos and Ajo-Franklin (2021); F21, Fokker et al. (2021)

In particular, we use fluid volume fractions of 0.6 and 0.2 for lacustrine sediments and all other materials, respectively, from Ortega and Farvolden (1989) as an approximation for porosity. Furthermore, we use approximate bulk moduli of 2.5 GPa and 35 GPa for the pore water and rock matrix, respectively. We use these values to estimate Skempton's B following Roeloffs (1988), and adopt their estimate of the undrained Poisson ratio. We conduct an MCMC inversion to determine the parameters $s_t$, $p_0$, $\Delta c$, $\tau_{\max}$, $a$, and $b$ using the Euclidean distance between the observed and modeled $\frac{dc}{c}$ with the `emcee` python package (Foreman-Mackey et al., 2013). Details of the initialization and convergence of the `emcee` runs are given in supplementary sect. 4. Median models are shown in Figure 5 for the lag window 4-8 s of the 2-4 Hz frequency band; results in terms of median models and model ranges for all frequency bands and lag windows are shown in Supplementary Fig.s S5 and S6.

## 5 Results and discussion

### 5.1 Comparison of co-located sensor results

Comparison between observed $\frac{dc}{c}$ at the co-located seismometer and accelerometer of the station G.UNM shows overall strong consistency (Fig. 4). Correlation coefficients between observed time series range from 0.83 to 0.99, with the exception of the East component at 0.5 Hz and 2.0 Hz, where a gap in highly coherent observations results in an offset of $\frac{dc}{c}$ between seismometer and accelerometer after the Puebla earthquake. Consequently, we discourage the analysis of single components; a synoptic analysis or weighted average of components as suggested, e.g., by Hobiger et al. (2014) is preferable. We conclude that strong-motion stations yield overall good results for $\frac{dc}{c}$ studies in urban settings with high-amplitude anthropogenic noise comparable to G.UNM.





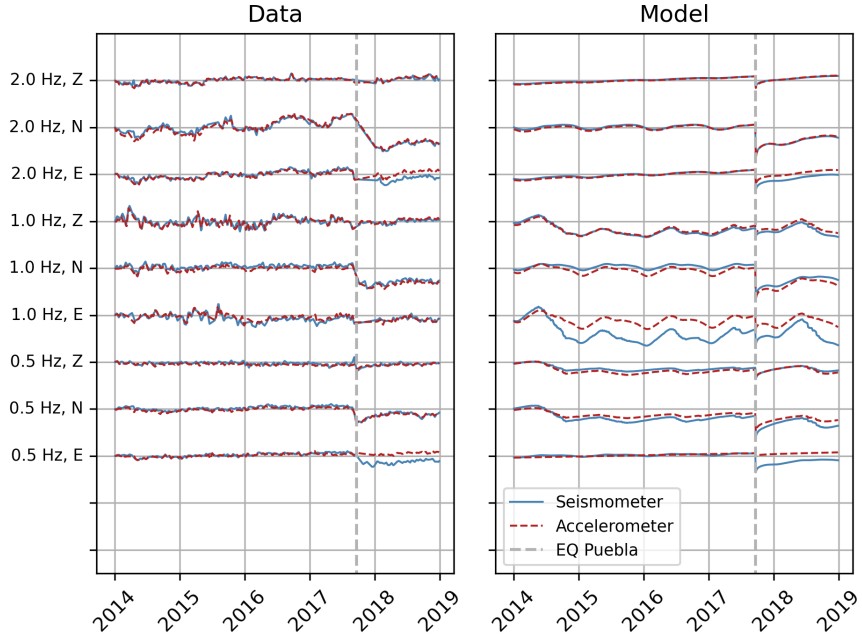

**Figure 4.** Comparison of $\frac{dc}{c}$ results co-located broad-band seismometer and accelerometer at station G.UNM. Left panel: Data for the duration of operation of both sensors (since 2013), for all components (E, N, Z) of autocorrelations, and all frequency bands (0.5-1 Hz, 1-2 Hz, 2-4 Hz and 4-8 Hz). Right panel: Median posterior models obtained by MCMC inversion, constrained by the data on the left. Both data and model are in good agreement between the sensors, with the exception of the East component.

## 5.2 Observed and modeled velocity changes for RSVM accelerometers and G.UNM seismometer

Results at 2-4 Hz for strong-motion stations of RSVM and the seismometer at G.UNM (Fig. 5) illustrate the seasonal $\frac{dc}{c}$ variations, and the drops and recoveries for the 2017 Puebla (19.9.2017) and 2020 Oaxaca (23.6.2020) earthquakes. Despite 280 its simplicity, our model captures the behavior of the observed velocity changes reasonably well. For RSVM and G.UNM at 2-4 Hz, lag window 4 – 10 seconds, the mean correlation coefficient (CC) between modeled and observed $\frac{dc}{c}$ is 0.77 and the median CC 0.84. (0.5–1.0 Hz: 0.68 / 0.75; for 1-2 Hz: 0.67 / 0.73; 4-8 Hz: 0.69 / 0.76). The inversion failed to converge in approximately 10% of cases, which we include nevertheless (see Supplementary Figs. S5 and S6). We exclude results above a misfit threshold set at the bottom quartile from further analysis. These models are shown by dashed white lines in Fig. 5.

Our model generally fits better for stations inside the basin, including transition zone and lake zone station, than those outside. This is reflected by a consistently higher CC between observed and modeled $\frac{dc}{c}$ for stations inside the basin, and a slightly lower average misfit.

We propose three reasons for the better performance of the model inside the basin. First, incident ambient noise may be more stable near the city center due to repetitive traffic patterns and a generally higher amplitude. A detailed analysis of noise source



conditions is out of scope of this work but may benefit future urban monitoring studies. Second, the assumption that obser-
vations are dominated by scattered surface waves may be more appropriate for basin stations located on stratified sediment
with strong impedance contrasts. Third, better *a priori* information is available about the basin subsurface structure compared
to areas outside of the basin thanks to the detailed thickness information of the low-$v_s$ aquitard (Solano-Rojas et al., 2015)
and to shallow profiles of $v_s$ from well logs (Singh et al., 1995, 1997). This results in likely more accurate station-specific

surface wave sensitivity kernels inside the basin, which illustrates the value of *a priori* site-specific geologic and hydrologic
information for quantitative shallow structure monitoring with ambient noise.

Several previous studies have applied detailed physics-based models to $\frac{dc}{c}$ , usually focusing on a small number of selected
stations (Tsai, 2011; Richter et al., 2014; Lecocq et al., 2017; Fokker et al., 2021; Illien et al., 2021, 2022) or time series
(Rodríguez Tribaldos and Ajo-Franklin, 2021). Previous array-wide analyses have mostly focused on empirical transfer func-

tions between hydrologic and meteorologic parameters and observed $\frac{dc}{c}$ (e.g. Wang et al., 2017; Donaldson et al., 2019). Here
we take a partially physics-based approach to the scale of a sedimentary basin and metropolitan seismic array, using a com-
prehensive set of time series in terms of frequency (0.5-8 Hz in octaves) and spatial components (E, N, Z), and integrating
multiple processes influencing $\frac{dc}{c}$ (poroelastic stress, thermal stress, non-linear elasticity with co-seismic drop and recovery).
It is straightforward to extend our approach to other surface wave modes (Love waves, higher modes). Here, we chose to limit

our analysis to fundamental-mode Rayleigh waves since limited information on subsurface structure is available and we have
limited knowledge of the surface-wave modal representation in scattered waves.

A current limitation of our model is that we assume a linear superposition of different processes affecting $\frac{dc}{c}$ (see Illien et al.,
2022, for a discussion on this point). Nevertheless, it is the first step to a comprehensive model for $\frac{dc}{c}$ based on simplified
physics at the basin scale.

## 5.3 Parameters controlling seasonal velocity variations

The parameters relevant to seasonal variations in our model are (i) thermal diffusivity, (ii) hydraulic diffusivity, (iii) the sen-
sitivity of $v_s$ to thermal stress, $s_t$ (which summarizes non-linear and linear elastic properties), and (iv) the minimum effective
pressure $p_0$. For practical reasons, only few values were tested for (i) and (ii). Therefore, we only present the inference of $s_t$
and $p_0$ by the inversion. We observe variability between the Z, N, and E components for both parameters, as well as between

the frequency bands. Depending on the station and frequency band, either temperature or precipitation emerges as the dominant
seasonal control from our inversion; this is apparent from results shown in the supplementary material.

In previous studies, single-station correlations have been interpreted as multiply scattered body waves (Richter et al., 2014;
Sánchez-Pastor et al., 2018, e.g.), whereas we interpret them as surface waves. This is supported by elliptical particle motions
(Supplementary Fig S7). The coda of cross-correlation is usually interpreted in terms of surface waves (Table 2).The main

difference between the auto- and cross-correlation coda is receiver separation, which may not suffice to change from observing
predominantly scattered surface waves to predominantly scattered body waves. Moreover, Yuan et al. (2021) found in numeri-
cal experiments that surface waves tend to dominate single-station correlation sensitivity in scattering media in the presence of
depth-varying velocity structure, which is the case in sedimentary basins. For these reasons, we consider an interpretation in





**Figure 5.** Time series of relative velocity change (circles) and median models (white lines) in the frequency band 2-4 Hz for autocorrelations of the East, North, and vertical components of RSVM stations and G.UNM. Stations are arranged by topographic elevation outside the basin and by aquitard thickness from Solano-Rojas et al. (2015) for stations inside the basin. Stations at lower elevation (near the basin edges) tend to be located on alluvial sediment, while those at high elevation are on bedrock. Color coding shows the aquitard depth in shades of yellow to black (minimum = 0 m, maximum = 75 m) and elevation in shades of yellow to green. Vertical red dashed lines indicate the timing of the 2017 and 2020 earthquakes, while the horizontal grey dashed line indicates the boundary between stations at hard sites and stations at intermediate and soft sites. Vertical grid spacing is 1% velocity change.

terms of surface waves sensible. Apart from the study by Yuan et al. (2021), the single-station correlation sensitivity to velocity
changes remains scarcely researched in comparison to cross-correlation coda sensitivity. This topic merits further attention.
Assuming that surface waves dominate our observations, we consider that the variation of parameter values with frequency
range is an effect of surface-wave dispersion. As a working hypothesis for differences of parameters between components, we
assume that waves recorded on E, N, and Z-component are dominated by different scattered arrivals or different wave modes





and therefore are sensitive to different parts of the surrounding medium.

### 5.3.1 Sensitivity of $v_s$ to thermal stress

Inverted values for the depth-independent sensitivity $s_t$ of $v_s$ to thermal stress range from $10^{-6}$ to 0.1 K$^{-1}$ (Fig. 6). Inside the basin, $s_t$ ranges from $10^{-6}$ to 0.01 K$^{-1}$. For the frequencies 1-2 and 2-4 Hz, we observe that stations with high sensitivity to surface temperature variations are mostly located in and near the lake zone, whereas lower $s_t$ is found mostly in the transition zone. This split disappears at 4-8 Hz.

When considering equation 3 and assuming that Poisson's ratio $\nu$ ranges approximately from 0.2 to 0.5, thermal expansion $\alpha$ from $10^{-6}$ to $10^{-5}$ K$^{-1}$, and elastic non-linearity $\frac{\partial \rho v^2}{\partial \sigma_c}$ from 10 to 1000 (Richter et al., 2014), then a physically reasonable range of $s_t$ is $3 \cdot 10^{-5}$ to $6 \cdot 10^{-2}$ K$^{-1}$. Most inferred values fall into this range. However, at hard sites (i.e. stations outside the transition zone outlined in yellow on the Map in Fig. 1), particularly at those at higher elevation, $s_t$ estimates are beyond what appears physically plausible. As discussed in sect. 5.2, we believe that the lack of knowledge of subsurface structure at these stations leads to inaccurate depth sensitivity kernels. In particular, near-surface sensitivity may be underestimated as no near-surface low-velocity layer is included in hard site profiles, which in turn could lead to an overestimate of the temperature sensivitity.

A second possible reason is the fact that we neglected the depth term of Berger's thermo-elastic solution. If we consider that surface wave sensitivity is greater at depths for sites where the medium does not have a thick low-velocity sediment cover, then it is plausible that this term is more important for bedrock sites than for soft sediment sites.

### 5.3.2 Minimum effective pressure $p_0$

Inverted effective pressures are mostly in the range of 0.01 Pa to 10 kPa. While the spatial pattern of the results varies, the stations with the highest minimum effective pressures are mostly found in and near the lake zone (Fig. 7).

Our observations may not constrain the details of the shallowest sediment, as surface waves may not be sensitive to the topmost layers at all sites and frequency ranges. Confining pressure may reach about 30 to 50 kPa in the first 20 m. Thus, inverted values of $p_0$ of up to 10 kPa are in a reasonable range.

Our current model of poro-elastic stress is based on a homogeneous halfspace solution (Roeloffs, 1988), whereas the central Mexico City basin is known to have a complex hydrologic structure with an aquitard and several underlying aquifers between interbedding of lacustrine sediments and tephra deposits (Arce et al., 2019). In addition, the built environment strongly influences whether water can enter the sediment; the surface rainwater is retained at the surface, or redirected in pipes. A more refined model of hydrological $\frac{dc}{c}$ could be constructed if measured hydraulic head of groundwater wells at high temporal resolution were available. The results of $p_0$ are nevertheless informative. Low values of $p_0$ (i.e. a high value on the right hand side of equation 4) suggest that $v_s$ is sensitive to precipitation at a site, while a high $p_0$ indicates the opposite.

Lower effective pressures in the hill and transition zones show that $v_s$ is sensitive to precipitation there. This may be because these sites are located where the volcanic and alluvial-pyroclastic aquifers are close to the surface (Vargas and Ortega-Guerrero,




2004). Two stations in the lake zone (MULU, VRVM, see Fig. 1) also show low $p_0$ at lower frequency of 0.5-1 Hz. A possible interpretation is that the lower-frequency observations possess some sensitivity to the shallow aquifer, which lies at a depth of approximately 50 m at those sites, while observations at 2-8 Hz are mostly sensitive to the lacustrine clay of the overlying impermeable aquitard.

Assi Hagmaier (2022) recently identified the presence of poroelastic seasonal velocity variations in the Valley of Mexico. Her analysis shows that these are too small to visibly influence the resonance period identified by horizontal-to-vertical spectral ratios (HVSR). However, she noted that an additional resonance peak dominates in the rainy season at lake zone sites, but disappears in the dry season. These findings stress that it is important to study seasonal effects on both $\frac{dv}{v}$ and HVSR, especially considering that the latter is commonly used for site effect assessment.

## 5.4 Co-seismic damage and recovery

We present the co-seismic drop and maximum relaxation timescale $\tau_{\max}$ of the 2017 Puebla earthquake with respect to peak ground acceleration (PGA) in Figure 8. Markers show median models of velocity drop and $\tau_{\max}$, error bars show the range between the 16th and 84th percentile. Not all strong-motion stations of the RSVM network recorded the Puebla earthquake; where PGA is not available from the RSVM stations, we use the PGA at the nearest available station (taken from horizontal ground motions), including the triggered strong-motion stations shown by black triangles in Figure 1. We add a dither (random shift) of up to 0.05 m/s$^2$ to the PGA values in order to ensure all markers are visible. Gray-shaded rectangles indicate values of $\tau_{\max}$ that lie beyond the duration of observation (approximately 2.5 years post-earthquake). Velocity drops strongly by up to $\approx$10%, and recovers slowly at most sites, with several $\tau_{\max}$ on the order of a decade. The base-10 logarithms of both $\tau_{\max}$ and the velocity drop show significant positive correlation with the logarithm of the PGA ($p < 0.05$), but the observed variances are not well explained ($R^2 = 0.28$ for $\tau_{\max}$ and $R^2 = 0.12$ for the velocity drop). We note that $\tau_{\max}$ is not very well constrained (wide bars).

Due to limited data coverage of the 2020 Oaxaca earthquake, we will not interpret results of its co-seismic velocity drop and recovery apart from stating that it caused a sudden drop in phase velocity at several sites in Mexico City which was smaller than the drop during the 2017 Puebla earthquake.

"Slow dynamics", the approximately logarithmic recovery of material mechanical properties after sudden changes induced by transient strains, is a subject of active research (TenCate, 2011; Sens-Schönfelder and Eulenfeld, 2019; Ostrovsky et al., 2019, e.g.). However, it is difficult to test current hypotheses in the metropolitan and geologically complex area of Mexico City given our sparse seismic network. To the extent that interpretation is possible, our results suggest that larger perturbations (in terms of higher PGA) may lead to a longer recovery time $\tau_{\max}$. This is consistent with other in-situ observations, notably Viens et al. (2018) who observed slower recovery of velocities in the Kanto basin after the 2011 Tohoku-Oki earthquake at sites that experienced higher strain rate during the mainshock. Illien et al. (2022) applied the Snieder et al. (2017) model to the 2015 Gorkha earthquake and aftershocks, and found a $\tau_{\max}$ of 846 days for the main shock, much longer than for the aftershocks (155 days). These observations run counter to laboratory experiments of slow dynamics, which suggest that the recovery of a particular material is independent of the perturbation amplitude (Shokouhi et al., 2017). The $\tau_{\max}$ on the order of 10 years inferred from





our study are uncharacteristically long compared to other studies, which reported 100 days for the Kanto basin (Viens et al., 2018), 250 days in Nepal (Illien et al., 2022), and 3 years in Parkfield (Brenguier et al., 2008a; Wu et al., 2016).

A possible explanation for the inferred long $\tau_{\max}$ is that the relaxation functions used to model $\frac{dc}{c}$ do not account for permanent damage. Permanent changes in the interferometric waveforms may lead to poor estimates of the relaxation time scale.

Permanent damage can be assessed using the decorrelation (Larose et al., 2010):

$$\text{Dcorr} = 1 - \text{CC}_i \tag{6}$$

in Figure 9. Here, $\text{CC}_i$ is the correlation coefficient of the stretched traces with respect to the average trace for 2017. Stations that were not operational during the 2017 Puebla earthquake are omitted and data with poor overall CCs are excluded (amounting to 40% of data at 0.5 Hz, 12% at 1 Hz, 3% at 2 Hz, 6% at 4 Hz). In the frequency bands 1–2, 2–4 and 4–8 Hz, decorrelation

increases after the time window containing the earthquake (gray rectangle). This increase is more pronounced for stations *not* located on the clay aquitard as defined in Solano-Rojas et al. (2015), although those stations experienced on average a lower PGA. At 0.5–1 Hz, decorrelation is less marked and is similar between stations on and off the aquitard. Using decorrelation as a proxy for permanent damage, this suggests that, compared to other sediments, the shallow clay of the lake zone would suffer less permanent damage.

Inferred seismic velocity drops in our model are high and weakly, but significantly, correlated in double-logarithmic space with PGA as well as with the strain rate proxy PGA / $v_{s10}$. We use the first 10 instead of the first 30 m here, as this distinguishes more accurately between hard sites and intermediate and soft sites. A dependence of the magnitude of the velocity change on peak ground motion amplitudes is expected based on laboratory studies (Shokouhi et al., 2017) and is consistent with earlier field studies (Viens et al., 2018).

## 5.5    Residual linear trend

Following the visual appearance of the $\frac{dc}{c}$ time series, we introduced a linear term in the $\frac{dc}{c}$ model (see sect. 4). The inversion results in strong slopes for this term with phase velocity increases of up to 0.75% per year (see Fig. 10). A linear term in a $\frac{dv}{v}$ model was previously used by Gassenmeier et al. (2016) who similarly introduced it on a heuristic basis as the data appeared to require it. Slope values were comparable to those we observe (0.27% per year) and were hypothesized to relate to recovery from

an earthquake that had occurred 5 years prior to the start of their observations with modified Mercalli intensity IV to V at the investigated site. Prior to the 2017 Puebla earthquake, the most recent earthquake that inflicted severe damage on Mexico City was the 1985 Michoacán earthquake (Singh et al., 1988a), which caused an MMI of IX in Mexico City (Arciniega-Ceballos et al., 2018). Despite its large intensity, we find it unlikely that the subsurface would still be recovering 30 years later (in a logarithmic regime, a recovery rate of 0.5% per year would require an initial velocity drop of at least 50%, which is unrealistic).

However, we cannot entirely rule out the possibility that the subsurface is recovering from the cumulative effect of multiple events.

We propose an alternative hypothesis. Mexico City has been undergoing rapid subsidence since more than a century due to groundwater extraction and sediment compaction (Solano-Rojas et al., 2015; Chaussard et al., 2021; Cigna and Tapete, 2021;



Cabral-Cano et al., 2008, and references therein). It has been suggested that another consequence of the subsidence process is
the reduction of the fundamental resonance periods of the sediment strata throughout the basin (Ovando-Shelley et al., 2007; Arroyo et al., 2013), which may be caused by an increase in seismic velocity as well as the compaction and resulting reduction of sediment strata thickness. Figure 10 shows the correlation between the slope of the linear trend of our models and the vertical displacement from InSAR, which we interpret as the effects of ground subsidence. Although the datasets cover different time ranges, subsidence rates have been approximately constant for decades (Chaussard et al., 2021) so that the rates can be directly
compared. Both rates are moderately and significantly correlated ($R^2 = 0.29$, $p < 0.0001$), indicating that sediment compaction may indeed be causing a velocity increase in Mexico City's underlying stratigraphy. Similar results from Taira et al. (2018) at the Salton Sea (California) showed a steady velocity increase at a much smaller rate of less than 0.1 %/year, which they interpreted as an effect of poroelastic contraction. Because of the importance of resonance frequency changes as estimated by Arroyo et al. (2013) for Mexico City's seismic hazard, the ongoing rapid subsidence and its associated hazard (Fernández-
Torres et al., 2022), and the magnitude of the changes, this topic merits further and more detailed investigation.

## 6   Conclusion

We presented a comprehensive study of seismic velocity changes in the Mexico City basin. Our study has several innovative aspects: i) We used autocorrelations of urban noise recorded by a strong-motion network, adapting the waveform clustering processing recently proposed by Viens and Iwata (2020). ii) We modeled array-wide velocity change time series using a linear
superposition of mostly physics-based terms, namely exponential relaxation for slow dynamics, poroelastic changes, thermoelastic changes, and a heuristic (not physics-based) linear trend. iii) We conducted a probabilistic inversion for the unknown model parameters.
We find that autocorrelations at strong-motion stations can be used for coda-wave monitoring at least in urban high-noise settings where results are comparable between a strong-motion and a co-located broadband sensor.
We estimated that observed velocity changes for frequencies above 2 Hz at soft sites atop very low-$v_s$ lacustrine sediments and at intermediate sites are mostly related to shear wave velocity changes in the top 100 meters, relevant to site effects. Our model performs best in this region of the array, likely due to the larger amount of prior knowledge on the shallow subsurface structure there. Observed seasonal velocity changes in this region and at these depths reach 1% peak-to-peak amplitude. At most sites, observed seasonal velocity changes show clear differences between East, North, and vertical components.
We showed that poroelastic and thermoelastic effects can be modeled in a self-consistent manner with physically reasonable results following Richter et al. (2014) and Roeloffs (1988) for stations inside the basin. Stations on thick lacustrine deposits show greater sensitivity to surface temperature than stations on shallow lacustrine deposits overlying the alluvial-pyroclastic aquifer. Sensitivity to precipitation appears to be greater for sites outside of the former lake perimeter on volcanic or alluvial-pyroclastic aquifers, while it is low atop the thick lacustrine deposit. Future research should investigate the spatial sensitivity
of different component autocorrelations, which are currently poorly understood in complex settings like sedimentary basins, so that differences between components which are now a nuisance may be turned into useful information.




Velocity drops during the Puebla 2017 and the Oaxaca 2020 earthquakes, followed by logarithmic recovery, indicate that sediments throughout the array show non-linear elastic behavior with transient strong velocity changes on the order of 1-10 %, supporting the conclusions by Singh et al. (1988b); Beresnev and Wen (1996) who stated that also the lacustrine sediment
behaves somewhat non-linearly. We conclude from stronger de-correlation at hard sites that permanent damage is more common there than on the soft lacustrine sediment sites. Future studies modeling slow dynamics using field measurements should account for permanent damage (e.g. through the added parameter of a static offset after the earthquake) and should investigate the cumulative effect of multiple earthquakes with longer-duration observations, in a similar way to Sawazaki et al. (2016).
Finally, we observe an increasing trend in surface-wave phase velocity that is positively and significantly correlated with ver-
tical displacements from InSAR, while InSAR measurements show the signal of compaction of the aquitard and aquifer due to groundwater extraction (Chaussard et al., 2021). As this trend is strong (up to approximately 1 %/year) and may provide additional depth-sensitive information about the compaction processes, it certainly merits further investigation also in light of reported resonance frequency changes by Ovando-Shelley et al. (2007) and Arroyo et al. (2013).

*Code and data availability.* Data from the Geoscope station G.UNM are public and can be obtained through FDSN web services
(doi:10.18715/GEOSCOPE.G). Data from the Red Sísmica del Valle de México are are collected and curated by the Mexican National Seismological Service (SSN). Seismic peak ground acceleration data at stations SCT2, CUP5, CCCS, and TACY were provided by the Accelerographic Network of the Institute of Engineering (RAII-UNAM), and are a product of the instrumentation, processing, and distribution of the Seismic Instrumentation Unit. The data is distributed through the Accelerographic Database System: https://aplicaciones.iingen.unam. mx/AcelerogramasRSM/. Topographic data used for shaded reliefs was obtained via PyGMT (Uieda et al., 2021) and is based on NASA
SRTM data (Farr et al., 2007).

Processing and correlation code `ants` is available at https://github.com/lermert/ants_2. The `NoisePy` module used for measuring stretching is available at https://github.com/Denolle-Lab/NoisePy. The Gaussian Mixture Model Clustering code of Viens and Iwata (2020) is available at https://github.com/lviens/2020_Clustering. The `ruido` python module used for data handling, urban noise clustering and stacking is available at https://github.com/lermert/ruido.


*Author contributions.* Conceptualization: MD, LE; Data curation: ECC, LQ, LE, EAFT, DMP; Formal Analysis: LE; Funding acquisition: MD(* supported LE), EAFT, ECC; Methodology and Investigation: LE, MD; Project administration: MD, ECC; Resources: MD, ECC, LQ; Software: LE, MD; Supervision: MD; Validation: LE, EC, ECC, DSR; Visualization: LE, ECC; Writing – original draft: LE; Writing – review and editing: LE, MD, EC, ECC, LQ.



*Competing interests.* The authors declare that they have no competing interests.

*Acknowledgements.* This research was enabled by continuous data provided by the RSVM (Red Sísmica del Valle de México) network. RSVM is maintained by the personnel of the Servicio Sismológico Nacional (SSN, Universidad Nacional Autónoma de México, Instituto de Geofísica). Financial support is provided by Consejo Nacional de Ciencia y Tecnología (CONACyT, National Council for Science and Technology) and the Government of Mexico City through agreements SECTEI/194/2017, CM-SECTEI/263/2021 and CM-SECTEI/156/2022.
Seismic peak ground acceleration data at stations SCT2, CUP5, CCCS and TACY were provided by the Accelerographic Network of the Institute of Engineering (RAII-UNAM), and are a product of the instrumentation, processing and distribution of the Seismic Instrumentation Unit. We gratefully acknowledge the data collected by Geoscope and the Tectonic Observatory, which are archived on IRIS. LE and MD have been supported by MD's fellowship from the David and Lucile Packard Foundation. ECC has been supported by DGAPA-PAPIIT project IN107321. EAFT is supported by a fellowship from CONACyT, México, under grant agreement CVU 863837. LE thanks the Pacific North-
west Seismic Network for supporting this research with caffeine, and Mouse Reusch and Paul Bodin for sharing their insights on telemetry gaps and the site conditions in the lake zone. We also acknowledge valuable discussions with Tim Clements about porolastic effects on seismic velocity, as well as with Naiara Korta Martiartu and Kurama Okubo about Monte Carlo inversions.



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



**Figure 6.** Inferred sensitivity of $v_s$ to thermal stress at seismic stations (Fig. 1). Colored triangles show the results for the North, vertical and East components at each station, where the inverted triangle indicates the station location and shows the value of the vertical component. White triangles show excluded models that had a misfit above the threshold. Grey triangles show the excluded observed data that did not pass quality control. Symbols for station AOVM are shown further east than the station to accommodate them on the map. Shaded relief from SRTM (Farr et al., 2007), geotechnical zonation from Gobierno de la Ciudad de México (2017).





**Figure 7.** Inferred minimum effective pressure $p_0$ at seismic stations (Fig. 1). Colored triangles show the results for the North, vertical and East components at each station, where the inverted triangle indicates the station location and shows the value of the vertical component. White triangles show excluded models that had a misfit above the threshold. Grey triangles show the excluded observed data that did not pass quality control. Symbols for station AOVM are shown further east than the station to accommodate them on the map. Shaded relief from SRTM (Farr et al., 2007), geotechnical zonation from Gobierno de la Ciudad de México (2017). Low values of $p_0$ correspond to a high sensitivity of $v_s$ to pore pressure changes, see equation 4.



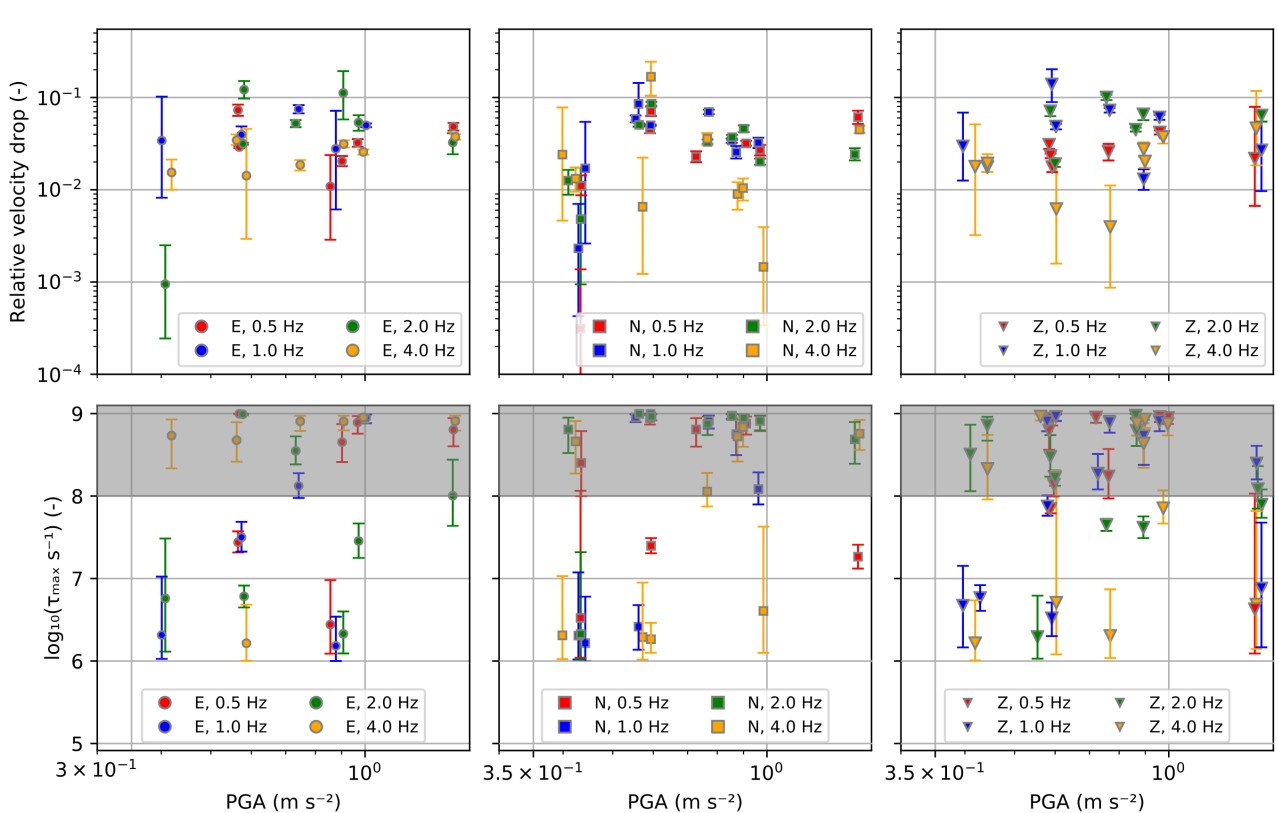

**Figure 8.** Inverted relative velocity drop and $\tau_{\mathrm{max}}$ for the 2017 Puebla earthquake. Top panels show the relative velocity drop for the E, N, and Z components. Bottom panels show the maximum recovery time scale $\tau_{\mathrm{max}}$ for the E, N, and Z components. The grey shaded area highlights inverted recovery timescales that surpass the recording duration. For both panels, error bars show the range between 16th and 84th percentile.

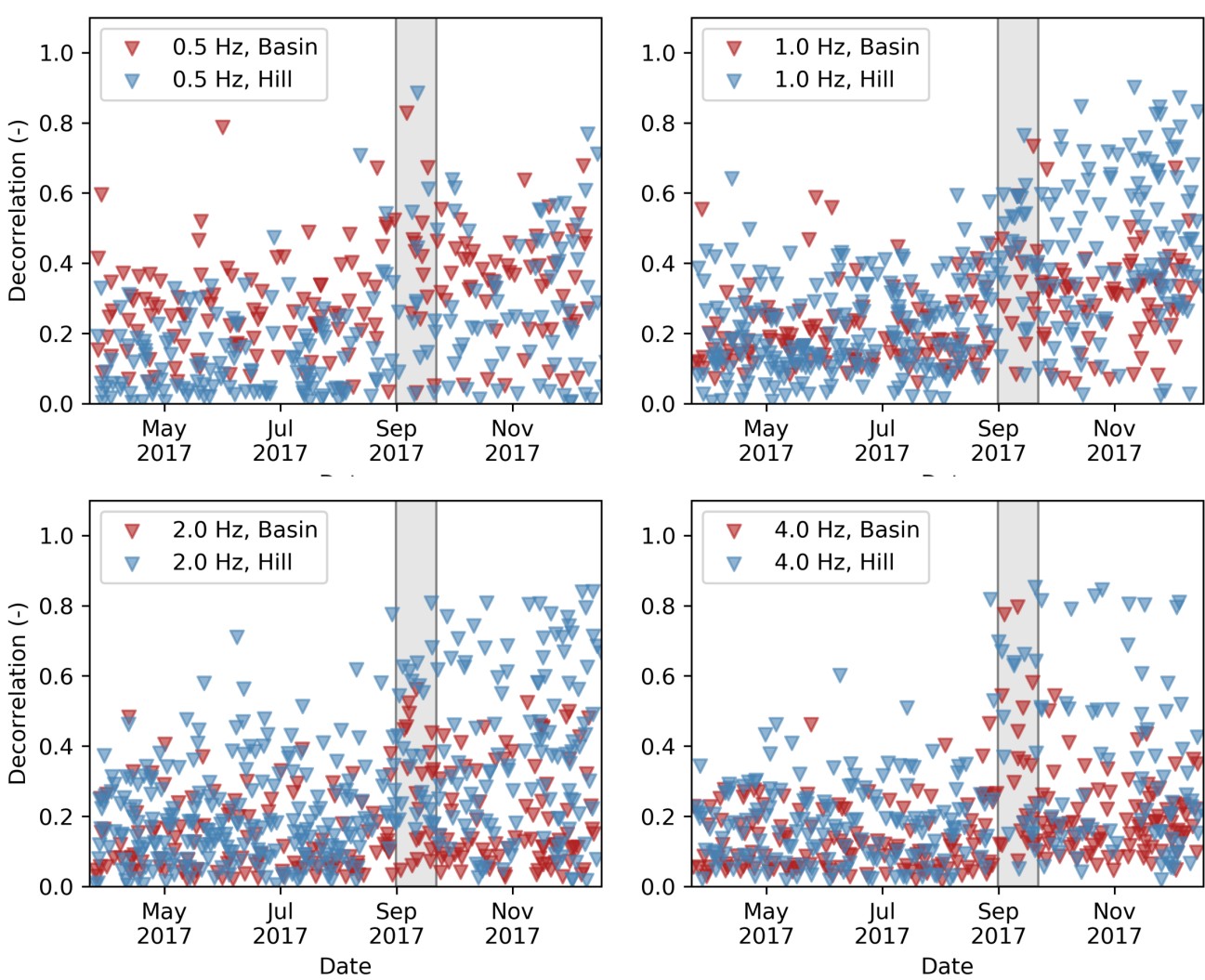

**Figure 9.** Observed decorrelation for stations located at sites with ("basin") and without ("hill") lacustrine sediments. The gray rectangle indicates the timing of the 2017 Puebla earthquake (time resolution of our stacks 20 days).





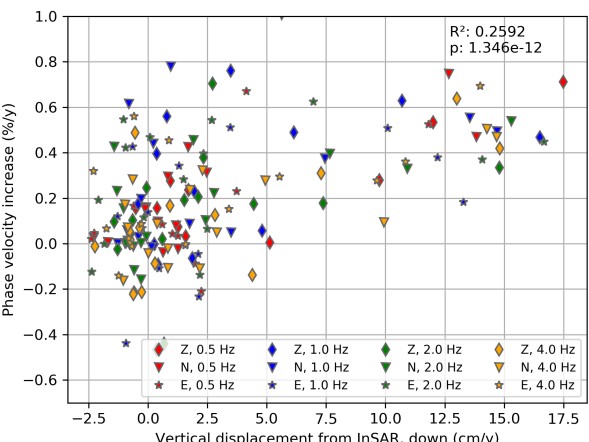

**Figure 10.** Slope of the linear term in the $\frac{dc}{c}$ model versus downward vertical displacement measured by InSAR. The moderate correlation suggests that the seismic velocity increase in Mexico City may be caused by compaction related to groundwater extraction.