# Peer review of "Probing environmental and tectonic changes underneath Mexico City with the urban seismic field"

_EGUsphere, 2022_

## Author Comment (AC2)

**EGUsphere, referee comment RC1**
https://doi.org/10.5194/egusphere-2022-1361-RC1, 2023
Comment on egusphere-2022-1361
Anonymous Referee #1
Referee comment on "Probing environmental and tectonic changes underneath Ciudad de México with the urban seismic field" by Laura Ermert et al., EGUsphere, https://doi.org/10.5194/egusphere-2022-1361-RC1, 2023

*This paper evaluates the seismic responses to environmental factors, long-term subsurface deformation, and tectonic events by analyzing the autocorrelation of continuously recorded urban seismic noise. The authors discussed these diverse effects, from observations to model and physical interpretation, as well as site-dependent local effects and the impact of different materials. In addition, multiple types of seismic stations are used to investigate changes in the shallow to intermediate subsurface medium. They demonstrated the reliability of utilizing strong motion sensors for monitoring local deformation.*
*In general, the graphs are clear and well-organized, and the interpretation and discussion of the results are reasonably comprehensive. I have no doubts about the methods, no additional comments on the discussion or conclusion, nor on the article's writing, which I believe is publication-worthy.*

We thank the reviewer for the positive comments. We were happy to read that our argumentation and discussion have convinced the reviewer.

**EGUsphere, referee comment RC2**
https://doi.org/10.5194/egusphere-2022-1361-RC2, 2023
Comment on egusphere-2022-1361
Alicia Hotovec-Ellis (Referee)

Referee comment on "Probing environmental and tectonic changes underneath Ciudad de México with the urban seismic field" by Laura Ermert et al., EGUsphere, https://doi.org/10.5194/egusphere-2022-1361-RC2, 2023

*In this paper, "Probing environmental and tectonic changes underneath Ciudad de México with the urban seismic field" by Ermert et al., the authors describe their efforts to observe and explain changes in the shallow seismic wave velocity across a city on highly variable near-surface geologic conditions. Overall, I found the paper to be a pleasure to read—equal parts rigorous and accessible. The authors are transparent with their methodology and assumptions, build on the wealth of previous observations (e.g., of shallow 1-D velocity structure to create site-specific sensitivity kernels) available to them, and have constructed a series of interpretations that testable and applicable elsewhere. I have no substantial comments on nor issues with the content of the paper itself. My only*

*comments are for the presentation (e.g., wording, figure edits), which I hope will incrementally improve an already excellent article.*

*Find below a few comments, with line numbers where applicable:*

*(General) – I found it a little jarring to go between the English/Español versions of place names. I think mostly it's the loss of the ´ in México that I notice most (which apparently was also dropped between the locations for 2 and 4 in the author list). It's a minor distraction, but I think it would be good to choose one name or the other (e.g., Ciudad de México or Mexico City) rather than use them interchangeably.*

We agree with the reviewer that it would have been better to choose a consistent name throughout. After discussing this point among the authors, we came to the conclusion that "Mexico City" is more appropriate. We have adapted the title and the manuscript accordingly.

*Line 51 – The phrase "not least due" was somewhat confusing to read the first time through. Perhaps "due largely" instead?*

We agree that this sentence was not very clearly formulated. We have reformulated it as (line 49/50):
"In the present study, we investigate both linear and non-linear changes of the seismic velocity underneath Mexico City. Mexico City has suffered devastating ground shaking, particularly due to the response of lacustrine clay deposits in the center  of the Mexico City basin [...]"

*Figure 1 – The color choices for this map are somewhat confusing. First, I think the color for the lake zone should be something other than blue. I recognize that blue for lake zone makes some sense, but I think it evokes too much that standing water still exists there (when it does not). Brown instead? The blue symbols for the Tectonic Observatory sites also blend too much into the lacustrine blue, so whatever choice is made for the lake zone, these symbols should be contrast to it. Also some color other than green should be used for UNM if possible, for those with red/green colorblindness to prevent confusion with the RSVM sties. Yellow? If you cite Crameri et al. (2020) in Figure 2, you should be sure to apply those concepts to the maps/other figures as well.*

We have gladly adopted the reviewer's suggestions. As requested by the journal, we had used the coblis colour blindness simulator (https://www.color-blindness.com/coblis-color-blindness-simulator/) prior to our first submission, and did not identify major issues in the main manuscript. However, this is only a simulator and we used it on figures that we already knew well. Therefore, it may not be representative of the experience of readers who see the figures for the first time. Moreover, we neglected to test it on the supplementary material. We are grateful that the reviewer pointed out this issue.
For Figure 1, we have now tried to improve the visibility of the G.UNM symbol by making it a bright color. To declutter the map, we have also removed the station labels of the triggered strong-motion stations (leaving only the symbols), as the station codes themselves are not relevant for the rest of the paper. We have adapted the caption accordingly.

*Figure 2 – I only see one cyan/black line when I print the figure on paper. I can see clearly when the lines are supposed to be without the lines themselves, so perhaps arrows at the top of the figure centered on the time would be enough to illustrate the dates of these changes?*

We thank the reviewer for pointing out the issues that the lines do not print well. We have simply followed the suggestion and added two arrows to mark the dates. We have adapted the figure caption accordingly:

"Top: Acceleration spectrogram of urban noise at G.UNM (North component), broadband seismometer. Prior to June 2008, the sensor was operated at lower gain. The spectrogram illustrates high noise levels characteristic for the urban location. Faint changes in the spectrum coincide with the September 2017 earthquake (left cyan line and arrow above the panel), as well as a marked drop in noise level during the Covid-19 pandemic (right cyan line and arrow: First announcement of anti-pandemic measures)."

*Line 114 – Obviously the cluster names for "day" and "night" are self-evident, but the difference between "noise" and "other" are not. Is "other" more like a transition between clear day and night? What constitutes "noise" in a way that it'd be correlated? Since you only end up using "day" during your work I don't think this is a major issue, but as I was reading, I did wonder about it.*

We thank the reviewer for pointing out that this is somewhat confusing. "Noise" is simply the smallest cluster, which at most stations appears to collect waveforms related to temporary situations. The "other" cluster is a consequence of using the most common optimal number of clusters (instead of forcing the number of clusters to 3). At many stations it indeed seems to fall in between the day and night clusters. We have added a corresponding short explanation to the manuscript (lines 113 ff):

"Here, "noise" refers to the smallest cluster of sporadically appearing, transient disturbances. The "other" cluster is required to fulfill the optimum number of 4 clusters. It mostly coincides with the transition between day- and nighttime."

*Figure 3 – Consider adding a color scale on the right side showing on the figure itself the color range and what it corresponds to.*

We have added a color scale showing the correlation coefficient of reference and current noise stack.

*Figure 5 – Similar comment as Figure 1, though I admit I do find the color mapping aesthetically pleasing despite the red/green split. I don't think this figure is too much of an issue as you do have a clear delimit between which sites are in or out of the basin, and they are ordered clearly by elevation.*

In this case, we had already debated about the color scale prior to submission. It appeared to us that the information is reasonably transmitted when using the color blindness simulator, but again we already knew what information to look out for. Even if the details of station elevation are not well captured by the color scale, they are contained in the order of the curves as the reviewer pointed out. We suggest keeping the figure in its current form.

*Lines 332-333 – You mention the full range of 10-6 to 0.1 K-1 back-to-back. Do you mean that basin sites had the full range of values that you observed? I just had to do a double-take that you hadn't repeated yourself here.*

Re-reading these sentences, we also noticed that they are slightly confusing. What we were trying to point out here is that the basin sites conform better to what we would expect than some of the hill sites; which may be among other things because the vs profiles we used there are more realistic. We summarized the two sentences into one, which hopefully makes reading easier (lines 332 ff):

"Inverted values for the depth-independent sensitivity $s_t$ of $v_s$ to thermal stress range from $10^{-6}$ to 0.1 K$^{-1}$ (all sites) and $10^{-6}$ to 0.01 K$^{-1}$ (only basin sites) (Fig. 6). "

*Line 379 – You mention velocity drops upward of 10% here. Is this right? The largest drops I see in Figure 5 are of order 1%. The scale on Figure 8 also only goes up to 100 and most sites do not approach even that. Please clarify if you mean that the 10% is referring to studies mentioned in the introduction (Lines 45-49) or if you observed this yourself, and if so where.*

Figure 8 (top row) showed the relative velocity drop, i.e. 0.1 corresponds to 10% and 1 corresponds to 100%. This is in fact confusing because we always express dv/v in % in the text and in other figures. We decided to adapt the figure accordingly (see also below, point about Fig. 8). The vertical axis of the upper panels is now in percent.

*Figures 6/7 – Only thought here is the lake sediment color again (these sites aren't on water!). I really like the presentation of the different components together like this!*

We adapted the lake zone color to brown, similar to the map in Fig. 1.

*Figure 8 – I'll admit I found this figure difficult to interpret, other than to show the variety of observations at different frequencies and on different channels. I'd suggest changing the colors here to be ordered by frequency (light to dark, or vice versa).*

We understand that the figure looks confusing. We have adapted the color scale of this figure and Fig. 10 according to the suggestion of the reviewer; we have also changed the axis of the upper panels in this Figure to % so as to be more consistent with the other figures and manuscript text.
Unfortunately, the figure remains hard to read. Ultimately, these two parameters are not very well constrained and show a high variability in between stations that makes the results hard to interpret in detail. What we can infer is that there is a slight positive correlation between velocity drop (in logarithmic scale) and PGA, consistent with other studies. We have made the following small adaptation to the manuscript text to point out more clearly the limitations of these results (lines 380 ff):
"We note that both τmax and, to a lesser extent, the velocity drop, are not very well constrained (wide bars)."

*Figure 10 – Same comment for color, though the main point of the figure is to illustrate the weak linear relation. I also found this figure oddly pixelated compared to the quality of*

*the other figures.*

We adapted the color scale to be ordered from dark to light. We also changed the legend arrangement to avoid covering data. Furthermore, we fixed a mistake: Previously, the linear regression was run after adding the dither. After correcting this (so that it is run directly on the results), the correlation coefficient is slightly increased.
We will double-check the resolution of the figure on the typesetting copy.

**Supplement**

*Supplementary Figure 1 – Please use the same color scheme as Figure 2 here.*

Thank you for pointing out this inconsistency. We adapted the color to the respective color scale, and dropped the years prior to 2014 from the raw cross-correlation results, as they are not relevant in this figure.

*Supplement Line 52 – 25'000 and 75'000 to 25,000 and 75,000 to be consistent with the use of commas in Line 49, or vice versa.*
We thank the reviewer for spotting this. Actually, the SI convention for separating numbers larger than 1000 is with a space, so we have now changed everything to be consistent with SI.

*Supplement Line 56 – Missing section number for "??".*
We apologize for this oversight, this reference did not work because the corresponding section is in the main manuscript; we have now added the section number and a note to point to the main manuscript.

*Supplementary Figure 2 – The dot-dashed red line is difficult to read. Consider instead a solid thin gray line or something similar instead?*
We have adapted the figure accordingly and agree it is more readable now.

*Supplementary Figure 3 – In the titles for these plots explicitly include which site type they are rather than having your reader infer it.*
We have added the site types to the titles of the figure.

*Supplementary Figure 4 – Same red/green line color issue.*
We agree with the reviewer, and we have attempted to mitigate the issue by a) choosing a different color and b) adapting the line styles of the curves to be different.

*Supplementary Figures 5-6 – There is a lot going on in these figures, but I very much appreciate the transparency in showing everything—including where the inversions didn't do a good job or converge. I printed these all out, and the figures are impossible to read if not on screen and zoomed in a lot. Is there a way to break these up to make them bigger when they are formatted into the supplement PDF? Also, same red/green choice.*

We thank the reviewer for pointing this out. Our intention is indeed to work along the ideal of transparent science. In this sense, it is important that the figure is readable. The only way we could think of to achieve this was to split each figure from the current supplement into 4

pages (for Supplementary Fig. 5) and 2 pages (for Supplementary Fig. 6). We hope that this will solve the issue.

*Supplementary Figure 7 – Is color here by time?*

Color is indeed by time. We have added a corresponding half-sentence to the figure caption.

*I look forward to seeing this paper in print, and hope you find my comments useful.*

---

## Author Response (AR2)

Manuscript EGUSPHERE-2022-1361 minor revision - reply to topical editor

We thank the topical editor for the friendly support in handling the manuscript and for providing additional comments. We have addressed them as follows:

*There are just some minor things to be corrected in the Abstract and Conclusions. In the abstract, please change "The subsurface material" into "The rocks beneath the Mexico City..." or something similar (sediments?).*

We have changed this formulation to "sediments underneath", as it is in fact mostly the sediments that are responsible for the strong site effects (together with the basin shape).

*In the Conclusions - in the first sentence, the formulation "in the Mexico City basin" is bit awkward, as there is no such geological formation. Maybe simply: beneath the Mexico City.*

We agree, previously we had not noticed this slightly odd formulation. We have changed it to "underneath Mexico City".

*Also note that currently the Conclusions are quite lengthy and contains also literature references. I would appreciate if you can make them shorter and avoid citing references in this section. All was already mentioned in the Discussion section.*

We thank the editor for this suggestion, which we think actually helped to make the conclusions more concise. We have removed all the literature references and shortened the text by removing some less essential details.
The references had indeed all been cited either in the methods or results&discussion part, with the exception of the references about non-linear behaviour of the basin sediments. Therefore, we added a new sentence to the results and discussion part, section 5.4: "Velocity decrease is observed across all the studied locations including those on lacustrine clay, supporting the conclusions by Singh et al. (1988b) and Beresnev and Wen (1996) who stated that the lacustrine sediment behaves non-linearly."

We will append a marked-up version of the changed manuscript and a clean version of the changed manuscript. No changes have been made to the supplementary material.